# Life Cycle Plasticity in *Typhula* and *Pistillaria* in the Arctic and the Temperate Zone

**DOI:** 10.3390/microorganisms11082028

**Published:** 2023-08-07

**Authors:** Tamotsu Hoshino, Yuka Yajima, Yosuke Degawa, Atsushi Kume, Oleg B. Tkachenko, Naoyuki Matsumoto

**Affiliations:** 1Bioproduction Research Institute, National Institute of Advanced Industrial Science and Technology (AIST), 2-17-2-1, Tsukisamu-higashi, Toyohira-ku, Sapporo 062-8517, Japan; 2Hachinohe Institute of Technology, Obiraki 88-1, Myo, Hachinohe 031-8501, Japan; 3National Institute of Polar Research, 10-3, Midori-cho, Tachikawa, Tokyo 190-8518, Japan; 4Graduate School of Engineering, Muroran Institute of Technology, Mizumoto-cho 27-1, Muroran 050-8585, Japan; y.yajima@mmm.muroran-it.ac.jp; 5Sugadaira Research Station, Mountain Science Center, University of Tsukuba, Sugadairakogen, 1278-294, Ueda 386-2204, Japan; degawa@sugadaira.tsukuba.ac.jp; 6Graduate School of Bioresource and Bioenvironmental Sciences, Kyushu University, Motooka 744, Nishi-ku, Fukuoka 819-0395, Japan; akume@agr.kyushu-u.ac.jp; 7N.V. Tsitsin Main Botanical Garden, Russian Academy of Sciences, 4 Botanicheskaya Ul., 127276 Moscow, Russia; ol-bor-tkach@yandex.ru; 8National Agriculture and Food Research Organization, Tsukuba 305-0856, Japan; nowmat@a011.broada.jp

**Keywords:** cryophilic, ecophysiology, growth temperature, life history, local climate, *Pistillaria petasitis*, *Typhula hyperborea*

## Abstract

*Typhula*ceae Jülich is one of the cold-adapted fungal families in basidiomycetes. The representative genera, *Typhula* (Pers.) Fr. and *Pistillaria* Fr., are distinguished by the discontinuity between stems and hymenia in the former and the continuity in the latter (Fries 1821). This taxonomic criterion is ambiguous, and consequently, the view of Karsten (1882) has been widely accepted: *Typhula* develops basidiomata from sclerotia, while basidiomata develop directly from substrata in *Pistillaris*. However, Corner (1970) observed basidiomata of *Pistillaria petasitis* S. Imai developing from sclerotia in Hokkaido, Japan. We later recognized that *P. petasitis* basidiomata also emerged directly from substrates on the ground in Hokkaido. An aberrant form of *Typhula hyperborea* H. Ekstr. was found in Upernavik, West Greenland. This specimen had a stem-like structure on a Poaceae plant, and sclerotia developed on its tip. Similar phenomena were found in other *Typhula* species in Japan. In this study, we aimed to elucidate the life cycle plasticity in the genera *Typhula* and *Pistillaria* through the interactions between their ecophysiological potential and environmental conditions in their localities. We collected and prepared strains of the above fungi from sclerotia or basidiomata, and we elucidated the taxonomical relationship and determined the physiological characteristics of our strains. Our findings imply that both *Typhula* and *Pistillaria* have the potential to produce sclerotia as well as the capacity for mycelial growth at ambient air temperatures in each locality where samples were collected. These findings suggest that *Typhula* spp. develope basidiomata not only from the sclerotia dispersed by the basidiospores but also from mycelia generated by the spore germination, which formed basidiomata multiple times, depending on their growth environments.

## 1. Introduction

Several kinds of microorganisms, especially fungi, have been reported in the cryosphere [1,2,3,4,5]. Fungal species were less frequently recorded from the cryosphere than those of the temperate zone, despite the fact that all fungal taxa have already been found in the cryosphere. These records suggest the possibility that various fungi are active in various cold environments. We proposed the term ‘cryophilic fungi’ to denote fungi adapted to the cryosphere [6]. The concept of cryophilic fungi is defined as fungi that are present in the cryosphere, spend a certain life stage or whole life cycle (sexual and/or asexual reproductive stages), and grow under subzero temperatures where water remains in the solid state, such as snow and ice. The concept of cryophilic fungi also applies to unculturable fungi, such as mycorrhizal fungi in snowbank fungi [7].

The life cycles of cryophilic fungi are affected by environmental factors [8]. This group of fungi includes psychrophiles and psychrotolerants. The psychrophily is defined by physiological characteristics at each stage of their life cycle [8]. Snow molds, representing cryophilic fungi, develop mycelia to attack dormant plants such as forage crops, winter cereals, and conifer seedlings under snow cover [5]. Some of them pass the dormancy from spring to autumn in the form of sclerotia in temperate and frigid zones, as well as the Arctic [9,10,11,12,13,14,15,16,17,18,19,20,21,22,23] and Antarctica [4,24,25], e.g., *Typhula incarnata* Lasch; *Typhula ishikariensis* complex (consisting of *T. ishikariensis* S. Imai; *T. canadensis* (J.D. Sm. and Årsvoll) Tam. Hoshino, T. Kasuya, and N. Matsumoto; and *T. hyperborea* H. Ekstr.); *Sclerotinia boreslis* Bubák and Vleugel.; and *Sclerotinia antarctica* Gamundí and Spinedi.

In 2007, the first author collected aberrant sclerotia of *Typhula* sp. (Figure 1D–F) from Upernavik (72.7869, −56.1472), West Greenland. They resembled those of *T. hyperborea* (Figure 1C) but were often formed on the tip of a stem-like structure on the Poaceae host. The stem-like structure was indistinguishable from the stems of *Typhula* sp. (Figure 1D–F) and seemed identical to the secondary sclerotia of hybrids of *T. ishikariensis* var. *ishikariensis* S. Imai × *T. ishikariensis* var. *idahoensis* (Remsberg) Årsvoll and J.D. Sm. produced under artificial conditions [26,27].

*Pistillaria* Fr. is close to the genus *Typhula* Fr. in *Typhula*ceae Jülich [28] and saprophytic. The hymenium of *Typhula* is distinguishable from the stem but indistinguishable from the stem in the genus *Pistillaria* [29]. However, this criterion proved unclear for separating both genera. Consequently, Karsten’s view has been widely accepted; *Typhula* spp. develop basidiomata from sclerotia, and *Pistillaris* spp., lacking sclerotia, develop basidiomata directly from substrata [30]. However, Corner [31] observed basidiomata of *Pistillaria petasitis* S. Imai developed from sclerotia in Hokkaido, Japan.

In this study, we aimed to elucidate the life cycle plasticity in the genera *Typhula* and *Pistillaria* through the interactions between their ecophysiological potential and environmental conditions in their localities.

## 2. Materials and Methods

### 2.1. Fungal Materials

Fungal sclerotia of *T. hyperborea* were collected from decayed leaves and stems of plants during the surveys. Sclerotium samples were packed in paper envelopes and dried at room temperature during transportation. Sclerotia were surface-sterilized in 70% (*v*/*v*) ethanol for 10 s, in 0.5% (as active chlorine) sodium hypochlorite solution for 30 s, and thoroughly rinsed in sterilized distilled water. They were then cut with sterilized steel blades, placed on potato dextrose agar (PDA: Difco, Sparks, MD, USA), and incubated at 4 °C. Mycelia from growing colony margins were transferred to PDA slants and maintained at 0 °C.

The collected basidiomata of *Typhula* sp. and *P. petasitis* were placed in plastic cases with wet cotton balls and kept in a refrigerator. Basidiomata were attached to the inside of Petri dish lids with double-sided adhesive tape, and spores were collected on PDA plates to incubate at 4 °C for 1 or 2 days. Basidiospores subsequently germinated and mated to produce heterokaryons and sclerotia. These sclerotia were transferred to fresh PDA plates and incubated at 4 °C for 2 weeks. Isolates were maintained on PDA slants at 0 °C.

These specimens were kept in the mycological herbaria of National Museum of Nature and Science, Tokyo (TNS).

### 2.2. Mating Experiments

Monokaryons of *T. ishikariensis* var. *ishikariensis* (strains PR7-6-7 and PR9-4-3 from Japan) and *T. canadensis* (strains 35-8 and 8-2 from Japan) were designated as testers and paired with dikaryons of collected strains (di-mon mating; [32]) on PDA plates and incubated at 4 °C for one month. A small agar block was cut from monokaryon colonies near the colony junction and transplanted to another PDA plate. Growth from the block was then examined for the presence of clamp connections on hyphae after incubation for 5 to 7 days at 4 °C. The presence of clamp connections on hyphae was the criterion of mating compatibility.

### 2.3. Phylogenic Analyses

Fungal strains were cultured for 1 month at 10 °C on PDA. Sclerotia of *T. hyperborea* from Upernavik, West Greenland, were harvested, and DNA was extracted by the protocol of DNeasy Plant MiniPrep (QIAGEN GmbH, Hilden, Germany). ITS regions, including the 5.8S gene of genomic rDNA, were amplified using primer pairs ITS1 (5′TCCGTAGGTGAACCTGCGG) and ITS4 (5′-TCCTCCGCTTATTGATATGC), as described by White et al. [33]. PCR products were purified using a QIAquick PCR Purification Kit (QIAGEN GmbH, Hilden, Germany) and sequenced in one direction on an ABI PRISM 3100 Genetic Analyzer (Applied Biosystems, Foster city, CA, USA) using the ITS1 primer. Multiple alignments of the ITS sequences were performed, and the nucleotide substitution rate (Knuc value) was calculated in CLUSTAL W [34]. A phylogenetic tree was constructed by the maximum likelihood method using the program MEGA X [35].

### 2.4. Morphological Observations

The colors of basidiocarps and sclerotia were described according to the color identification chart of the Royal Botanic Garden Edinburgh (Flora of British Fungi) [36].

For light microscope sections, aberrant sclerotia of *Typhula* sp. were fixed with 2% glutaraldehyde (Nisshin EM, Co., Ltd., Tokyo, Japan) in 50 mM phosphate buffer and washed in the same buffer. The samples were then post-fixed with 1% osmium tetroxide (Nisshin EM, Co., Ltd., Tokyo, Japan), dehydrated with an ethanol series, and embedded in Quetol 651 (Nisshin EM, Co., Ltd., Tokyo, Japan). The sections (0.8 μm thick) were stained with toluidine blue (Wako Ltd., Osaka, Japan) and observed under a light microscope.

### 2.5. Mycelial Growth Temperature

Mycelial discs of 5 mm diam were cut from the margins of actively growing colonies, inoculated to the centers of PDA plates, and incubated at six different temperatures from 0 to 25 °C in duplicate. After 1, 2, and 3 weeks of inoculation, colony diameters were determined. The linear mycelial growth rate per week was calculated after the initial lag period.

## 3. Results

### 3.1. Typhula hyperborea in Greenland

Fungal cultures from aberrant sclerotia from Upernavik, West Greenland, had feather-like mycelia when grown on PDA at 4 °C (Figure 2A), which were typical physiological reactions of *T. hyperborea* [23], and our strain did not react when paired with both tester monokaryons of *T. ishikariensis* var. *ishikariensis* and *T. canadensis*. Phylogenic analysis of ITS regions also supported this assumption (Appendix A).

When the cultures were kept at 4 °C for 2 years, sclerotia were found to have stem-like structures with intercalary, small secondary sclerotia, as described by Christen [26,27] (Figure 2B). Cultures of *T. ishikariensis* var. *ishikariensis* from Hokkaido, Japan, also produced similar aberrant sclerotia. Stem-like structures and aberrant sclerotia of *T. hyperborea* and *T. ishikariensis* var. *ishikariensis* on PDA were dark brown (19 bay) as the same color with field samples (Figure 1D,E). However, the normal stems of both fungi were white (2 B) to pale yellow (6 E) or pale brown (30 clay pink) [23]. Collected stem-like structures of *T. hyperborea* from Upernavik, West Greenland, were gradually connected with aberrant sclerotia (Figure 1F), and similar findings were reported at the base of the stem of *Typhula sclerotioides* (Pers.) Fr. at its origin from the sclerotium [37]. These results suggested that the germination of sclerotia and secondary sclerotium formation were reversible for *T. hyperborea* and *T. ishikariensis* var. *ishikariensis*.

In 1999, *T. hyperborea* OUP1811 was collected in Nuuk (64.1666, −57.7500) in West Greenland [23]. This strain also showed different culture morphology on PDA plates among the stains of *T. hyperborea* (Figure 2C). This strain had weak pathogenicity against the host [23], and they showed normal growth at 4–10 °C and abundant aerial mycelia without sclerotia on PDA. Sclerotium formation was, however, observed on oatmeal agar plates (Figure 2D).

### 3.2. Typhula sp. and Pistillaria Petasitis in Hokkaido, Japan

Basidiomata of *Typhula* sp., whose hymenium was distinguishable from the stem, were found on the dead petioles of *Kalopananax septemlobus* in the Ashoro Research Forest, Kyushu University, Hokkaido, Japan (43.2507, 143.5498, altitude 114–471 m) in October 2010 (Figure 3A,B). Basidioma: ca. 1.5–3.0 cm high, white (2 B). Head: ca. 2.5–8.5 × 0.5–1.8 mm, clavate to cylindric, obtuse, straight, or curved. Stem: ca. 3.5–21.5 × 0.5–1.0 mm, opaque, white (2 B). Although the specimens lacked sclerotia, the resultant cultures produced sclerotia on PDA (Figure 3C). Sclerotium: ca. 1–2.5 mm diam, globose to subglobose, almost black (36 fuscous black).

Basidiomata of *P. petasitis* without sclerotia were observed on the hillside of Mt. Asahidake (43.6511, 142.7990, altitude 1100 m), Higashikawa, Hokkaido (Figure 3A and Figure 4A,B), in August 2010. Basidioma: ca. 1–3.8 cm high, white (Figure 2B). Head: ca. 2.5–26.5 × 1.5–6.5 mm, clavate to cylindric, obtuse, straight, or curved. Stem: ca. 5–12.5 × 0.8–3.5 mm, opaque, white (Figure 2B). The fungus was also observed in the forest neighboring an agricultural field at the Hokkaido Agricultural Research Center, NARO, Sapporo, Hokkaido, Japan, in September 2010.

*Pistillaria petasitis* basidiomata were found on various substrates such as *Petasites japonicus*, *Conioselinum filicinum*, and *Cirsium kamtschaticum*. Basidiomata of *P. petasitis* were found on substrates on the ground surface up to 10 cm on 3 August 2010 (Figure 4D). Up to 80 cm on 23 August 2010 (Figure 4E). All *P. petasitis* isolates produced sclerotia under culture conditions (Figure 4C). The sclerotia of our isolates were similar to those described by Corner [37], ca. 2.5–22.5 × 2.5–5.5 mm, fusiform, somewhat flattened, and light brown (15 brick).

### 3.3. Effect of Temperature on Mycelial Growth

Optimal mycelial growth of *T. hyperborea* from Upernavik, West Greenland, *Typhula* sp. from Ashoro, and *P. petasitis* from Higashikawa occurred at 5, 15, and 20 °C, respectively (Figure 5). Maximal growth temperatures were 15, 25, and 30 °C, respectively. The mycelial growth range of *T. hyperborea* in Upernavik, West Greenland, was psyhrophilic and typical of this fungus [23]. According to the ranges of their mycelial growth temperatures, *Typhula* sp. from Ashoro and *P. petasitis* from Higashikawa were psychrotolerant. This is the first record of the mycelial growth range of *P. petasitis*.

These growth temperature relations agreed with the ambient temperatures when basidioma samples were collected in each locality. These results suggested that the mycelia of these three fungi could potentially and be active without snow in their localities.

## 4. Discussion

Snow molds representing cryophiles resume growth typically by carpogenic germination of sclerotia (sexual cycle) in autumn. Mycelia prevail on dormant plants under the snow to produce sclerotia in late winter before dormancy (asexual cycle). The life cycle of the genus *Typhula* is illustrated in Figure 6. Basidiospores germinate to develop into monokaryons, which subsequently mate with their counterparts differing in mating incompatibility alleles to produce dikaryons. Dikaryons are capable of sexual recombination through carpogenic germination of sclerotia, which is critical to generating diversity to cope with fluctuating environments and flexibility, as we found in this study.

Kawakami et al. [38] elucidated the conditions required for stem elongation from the sclerotium and fertile head development in *T. ishikariensis*. Stem elongation occurred at low temperatures and high humidity, but light was not essential. In contrast, light and moderate day length (8 h/day) were prerequisite for fertile head development. Several strains of *T. hyperborea* also produced basidiomata under Kawakami’s conditions [23]. *T. hyperborea* in the Arctic also acts to produce the asexual formation of basidiomata dispersing basidiospores under light conditions (red dashed line in Figure 6). Mycelia of *T. hyperborea* did not produce basidiomata under snow or dark conditions. *T. ishikariensis* complex formed sclerotium on the top of the remaining stems (Figure 1D,E). Our observation of sclerotia with stems was similar to secondary sclerotia described by Christen [26,27].

In addition, Tkachenko [39] reported another type of secondary sclerotia in *T. ishikariensis* var. *ishikariensis* on tulip bulbs in Russia. One to seven secondary sclerotia were found within original sclerotia. We also observed this type of secondary sclerotia from bloated original sclerotia from *T. hyperborea* from Upernavik (Figure 2B) and other strains of the *T. ishikariensis* complex (Hoshino et al., unpublished results).

Corner [37] suggested that the aggregation of hyphae of *Typhula gyrans* (Batsch) Fr. developed into the tissue of the stem and sclerotium. Hyphal aggregation of *Macrotyphula phacorrhiza* (Reichard) Olariaga, Huhtinen, Læssøe, J.H. Petersen, and K. Hansen also extended throughout the head. However, the texture of the hyphal walls in the head and central part of the stem was never as tough as in the sclerotium or on the surface of the stem. In addition, the mycelia of *M. phacorrhiza* and other *Macrotyphula* spp. on PDA plates were stroger than those of *Typhula* spp. (Hoshino et al., unpublished results). *M. phacorrhiza* was the type species of *Typhula*. However, it has presently been changed to *T. incarnata* [40]. These points suggested that stem and sclerotium had common properties in *Typhula* and related genera. According to Lind [41], rusts, smuts, and species of Dothideales Lindau in the Arctic have perennial mycelia in the host, enabling them to grow as soon as the season starts. Previously, we reported that several strains of *T. hyperborea* in West Greenland had weak pathogenic activity [23] and abundant aerial mycelium with less productivity of sclerotium (Figure 2C,D). These physiological characteristics supported adaptation to the Arctic summer climate (blight and cold conditions).

Similar phenomena were also recorded from other *Typhula* sp. in the Arctic (loss of sclerotium-forming ability under cultural conditions) [22] and *T.* cf. *subvariavilis* in Antarctica (no sclerotia at the field survey) [25]. Basidiomata of *T.* cf. *subvariavilis* in Antarctica emerged directly from substrates, and this fungus had high homology in the ITS region with *Typhula* sp. Wh-1 in Iran [42] and *Typhula variabilis* Riess, which is rather ubiquitous in the Northern Hemisphere, including areas with rare snow cover such as the Azores [43]. Snow mold symptoms of *T.* cf. *subvariavilis* in Antarctica and *Typhula* sp. Wh-1 in Iran were not observed in sclerotia just after the snow melts.

Sclerotia of *Typhula* sp. in Iran were formed ca. 2 weeks to 1 month after the snowmelt (Figure 7), remained immature as mycelial aggregations after snow melt, and matured without snow. *T. canadensis* in Norway also had sclerotia with aerial mycelia in the field, which were considered to facilitate dispersal by the wind [44,45]; however, they matured under the snow cover. Therefore, it was different from the ecophysiological characteristics of sclerotia from *T.* cf. *subvariavilis* in Antarctica and *T. canadensis* in Norway. These results suggested that *T.* cf. *subvariavilis* in Antarctica and other related species could still act after the snow melts in their localities.

Most of the *T. hyperborea* strains showed irregular growth on PDA at more than 10 °C [23]. However, these strains showed normal growth at the same temperature on corn meal agar or PDA with free radical scavengers such as ascorbic acid or β-carotene [23]. When *T. ishikariensis* complex, *T. incarnata*, and *Typhula trifolii* Rostr. from Canada were first exposed to the maximum growth temperature (20 or 25 °C) and then incubated at their optimal growth temperatures, the *T. ishikariensis* complex formed a “fan-like” irregular colony that was similar to the colony morphology of typical *T. hyperborea* [46]. Oxygen uptake of the *T. ishikariensis* complex was optimal at 20 °C (maximum growth temperature), about 15 °C higher than its optimal growth temperature (5 to 10 °C). Typical strains of *T. hyperborea* have strong pathogenicity against host plants. Therefore, they obtained free radical scavengers from hosts.

On the other hand, several strains of *T. hyperborea* from West Greenland obtained saprophytic activity (we did not collect such strains from East Greenland). Probably, they lost pathogenicity and acquired resistance to oxidative stress near their maximum growth temperature. There is a positive correlation between virulence and psychrophily. However, this significance is unclear [47,48]. Most species belonging to *Typhula*ceae are psychrotolerant and saprophytic. The pathogenic species of *Typhula* spp. found the new resource of overwintering plants and evolved in a cold environment under snow cover.

Many rusts in the Arctic have a simplified life cycle, only producing one kind of spore (micropuccinia) instead of a life cycle with three spore forms (eupuccinia) more commonly seen in warmer areas [48,49]. It is the first finding that the new life cycle stage of the *T. ishikariensis* complex and similar phenomena were observed in *Typhula* sp. in Ashoro, Hokkaido (Figure 3) and *P. petasitis* in Higashikawa and Sapporo, Hokkaido (Figure 4). Many types of research on the *T. ishikariensis* complex were carried out in temperate or frigid zones where air temperatures were higher than those of their psychrophily. Therefore, dikaryons of the *T. ishikariensis* complex grew only under the snow cover and formed sclerotia for the passing spring and summer seasons.

## Figures and Tables

**Figure 1 microorganisms-11-02028-f001:**
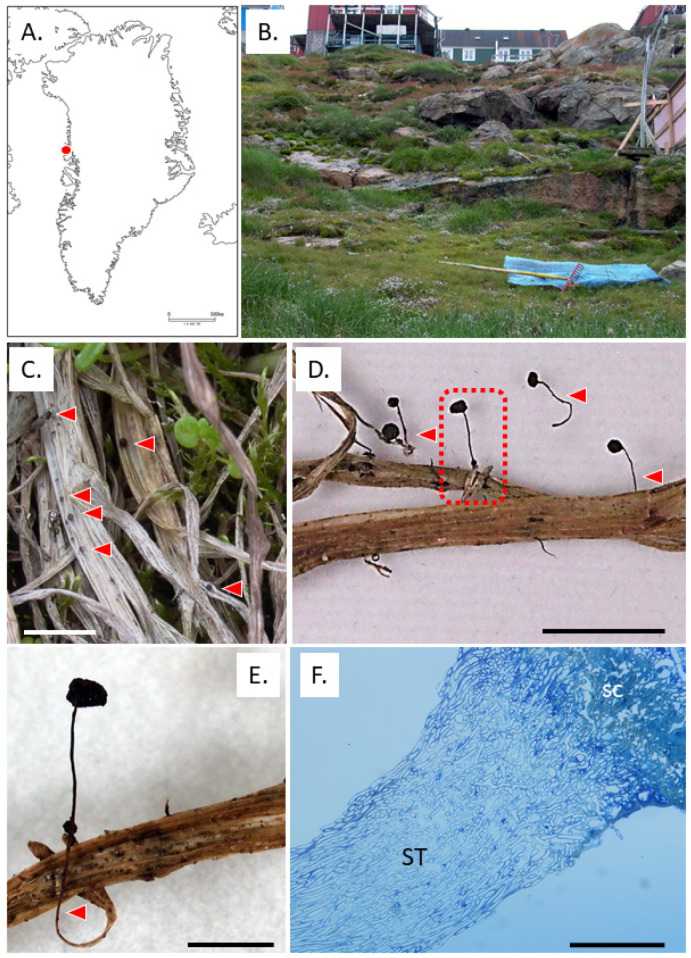
Aberrant sclerotia from Upernavik, West Greenland. Collected locality (**A**). Collected site (**B**). Normal sclerotia of *T. hyperborea* (**C**). Collected aberrant sclerotia (**D**). Red triangles: stem-like structures. Close-up view of the rectangular in (**D**,**E**). Vertical section of aberrant sclerotia (**F**). SC: sclerotia, ST: stem-like structure. Bars: 1 cm (**C**–**E**) and 100 μm (**F**).

**Figure 2 microorganisms-11-02028-f002:**
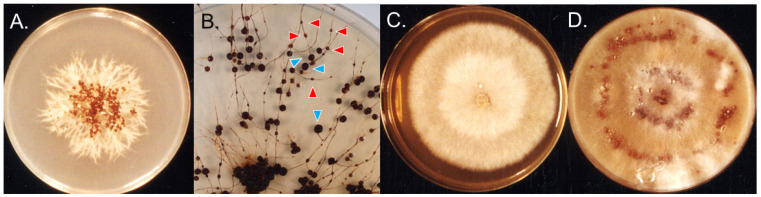
Mycelial growth of *Typhula hyperborea* from Greenland. Isolate from irregular sclerotia from Upernavik on PDA at 4 °C for 1 month (**A**) and 2 years (**B**). Blue triangles: original sclerotia. Red triangles: secondary sclerotia. OUP1811 from Nuuk on PDA at 4 °C for 1 month (**C**) and oatmeal agar plates at 4 °C for 1 month (**D**).

**Figure 3 microorganisms-11-02028-f003:**
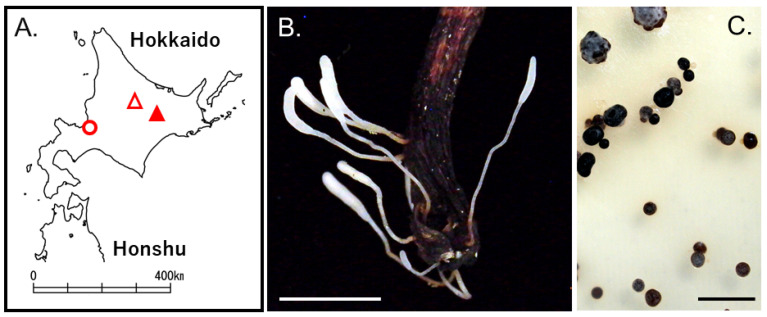
*Typhula* sp. on *Kalopananax septemlobus* in Ashoro, Hokkaido, Japan. Collected locality (**A**). Closed red triangle: Ashoro Research Forest, Kyushu University; open red triangle: Mt. Asahidake, Higashikawa; and red circle: Sapporo. Basitiomata of *Typhula* sp. (**B**). Sclerotia of *Typhula* sp. on PDA at 4 °C for 1 month (**C**). Bars: 1 cm (**B**) and 5 mm (**C**).

**Figure 4 microorganisms-11-02028-f004:**
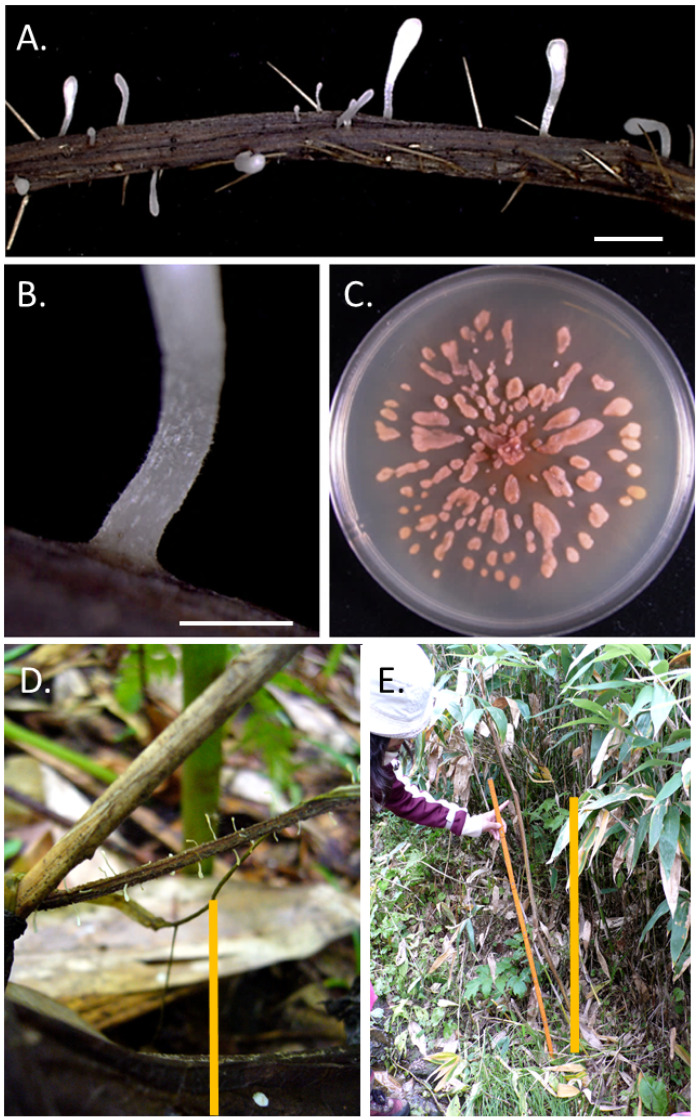
*Pistillaria petasitis* on Mt. Asahidake, Higashikawa, Hokkaido, Japan. Basitiomata of *P. petasitis* (**A**,**B**). Sclerotia of *P. petasitis* on PDA at 4 °C for 1 month (**C**). Field observation of heights of basidioma on 3rd August (**D**) and 23rd August in 2010 (**E**). Bars: 2 cm (**A**), 1 cm (**B**), 5 mm (**C**), 10 cm (**D**), and 80 cm (**E**).

**Figure 5 microorganisms-11-02028-f005:**
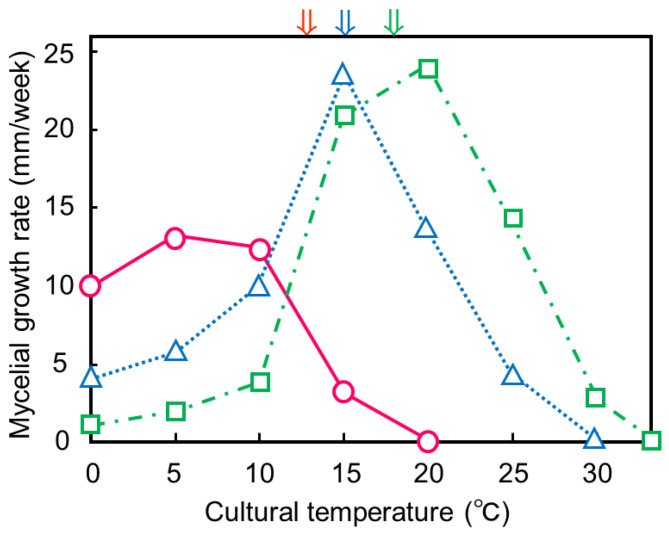
Effect of cultural temperature on mycelial growth. Red circles: *Typhula hyperborea* in Upernavik, West Greenland. Blue triangles: *Typhula* sp. on Ashoro, Hokkaido, Japan. Green squares: *Pistillaria petasitis* on Mt. Asahidake, Higashikawa, Hokkaido, Japan. Red arrow: 12 °C, maximal air temperature in Upernavik. Blue arrow: 15.5 °C, average air temperature in September in Ashoro. Green arrow: 18 °C, average air temperature in September in Sapporo.

**Figure 6 microorganisms-11-02028-f006:**
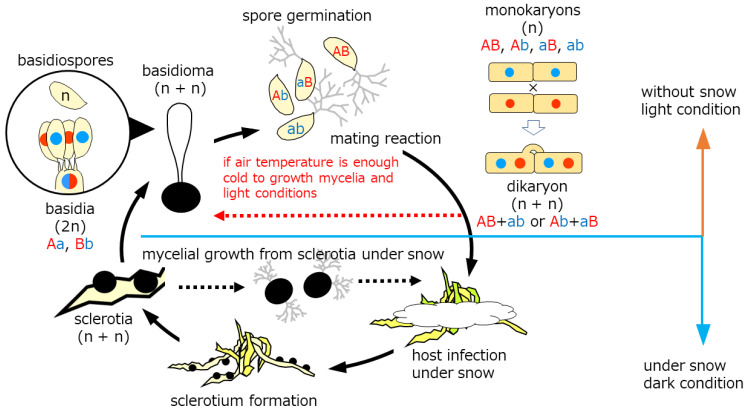
Life cycle of *Typhula* spp. in the cryosphere. Solid lines: sexual reproduction stages. Black dashed lines: known asexual reproduction stages. Red dashed line: our finding stage.

**Figure 7 microorganisms-11-02028-f007:**
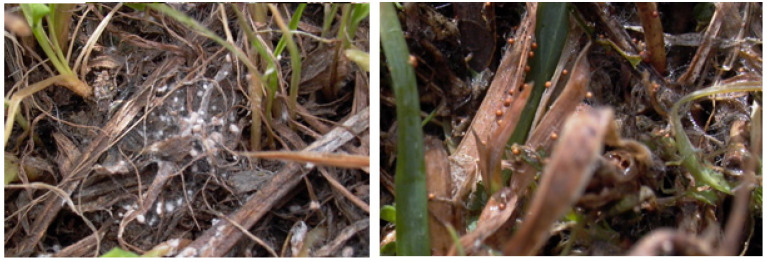
Sclerotia of *Typhula* sp. in northern Iran. Immature sclerotia, mainly mycelial aggregations, ca. 2 weeks after snow melt in Nir (38.0360, 48.0105), Ardabil Province on 27 February 2004 (**left**). Mature sclerotia, ca. 1 month after snow melt in Qazvin (36.2311, 49.9982), Qazvin Province, on 7 March 2004 (**right**).

## Data Availability

T.H., Y.Y., Y.D. and A.K.: Abstracts of 55th Annual Meetings of Mycological Society of Japan (https://doi.org/10.11556/msj7abst.55.0.19.0, accessed on 4 August 2023) in Japanese. O.B.T. and I.A. Schanzer: гриб *Sclerotium nivale* Elenev является аскoмицетoм (Ascomycete, *Sclerotium nivale* Elenev), (in Russian), 4 Съезд микoлoгoв Рoссии (The 4th Congress of Russian Mycologist), 12–14 April, Moscow, Russia.

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
