# Peer review of "Life Cycle Plasticity in Typhula and Pistillaria in the Arctic and the Temperate Zone"

_microorganisms, 2023, doi:10.3390/microorganisms11082028_

Round 1

Reviewer 1 Report

The MS investigates the plasticity in the life cycles of  Typhula and Pistillaria in the Arctic and temperate zones. Previous studies on the same, concluded that species belonging to those two genera differ in their life cycle as one was assumed to produce basidiomata from sclerotia, and in the other the basidiomata were developed directly from substrata, i.e. from the mycelium. .  However, the authors have encountered situations where fungi of these genera appeared in the other way in nature, thus raising questions on the certainty of the previous definition. In this MS, the authors describe some characteristics of the recently found mushrooms of species belonging to those tow genera in their natural habitat, as well as several developmental characteristics of the lab culture of the isolates, suggesting that members of both genera can produce basidiomata from the mycelial fraction, depending on environmental conditions. 

The findings are relevant in that they showcase the plasticity of basidiomycetes fungi in general, to adapt fruiting body production  according to environmental conditions, and more specifically in the cold region, where fungi are obligated to adjust to extreme temperature conditions.

However, the MS needs to be improved.

Below are several points for improvement:

General:

The MS needs careful perusal and correction by the authors. Several words seem to me to be computer-generated automatic characters and must be replaced.

For example: Line 157- “stme”-  to change to “ stem”,  Line 177- “secondary triangles” to change to “secondary sclerotia” and more.

Abstract:

1. Please add the aim of the study and methods. In my opinion the abstract contains only the introduction, description of the appearance of a new natural population and conclusion. Please add a description of the lab experiments molecular methods, lab studies on the effect of environmental conditions on growth rate and sclerotia formation etc., as well as the results that enabled your conclusions.

2. In line 24 : and consequently the view of Karsten (1882) has been widely accepted; Ty phula develops basidiomata from sclerotia”  please add   :” instead of “;”.

 Keywords:

 Please mention the names of the fungi studied:  Typhula hyperborean  and Pistillaria petasitis

Introduction:

1.       The introduction is too long and has basic information on the fungal world. It can be made concise by focusing on the topic of the paper.

2. Line 56: Please explain why mycorrhizal fungi have been defined as cryophilic fungi.

3. Line 60 starts with “Their life cycles”. Whose life cycles? General fungi, or cryophilic fungi? All fungi are affected by environmental factors.

4. Line 84- “In this study, we aimed to elucidate the plasticity in the genera Typhula and Pistillaria”. Please define the plasticity you aim to elucidate, whether “ life cycle plasticity” or another similar sentence.

5. Line 86- This is summarizing a conclusion that remains to be proved.

Materials and methods:

1. Please add the names of the studied fungi in the first section "Fungal material". I find it confusing that several fungi have been studied, with different names appearing in different sections. I could not find the outline. How many are important and how many are just for comparison?

2. Molecular identification.  Which of the fungi were identified by this method in this paper? All? Only the newly 2 isolated mushrooms?  Please add.

Results

Line 177: Fig 2- Replace the word triangles in  " secondary triangles" to " secondary sclerotia"

Line 196- the isolate “ strain OUP1811” appears here for the first time. Please describe the genus, history and reasons for adding it here.

Figure 4 E.  The fungi are not visible. You could change the legend of this figure to fit the picture. It appears in the same place as "D", but zoomed- out.

Figure 5 line 232: Please explain why the data are of the month of September, while the mushrooms were collected in August.

Conclusion: This section seems to be in part, a continuation of the result section with more investigations and results. Moreover, the discussion seems to be an extended review.  I suggest to shorten this section and to move part of the discussion to another review article.

Need carful proof reading and correction by the authors. Several words seem to me to be computer-generated automatic characters and must be replaced.

For example: Line 157- “stme”-  to change to “ stem”,  Line 177- “secondary triangles” to change to “secondary sclerotia” and more.

A

Author Response

         Thank you so much for the comments from the two reviewers. I am sending herewith our revised manuscript. Our incorporation of reviewers suggestions is as follows:

We are grateful to reviewer 1 for the critical comments and useful suggestions that have helped us to improve our paper, As indicated in the responses that follow, we have taken all these comments and suggestions into account in the revised version of our manuscript.

Comment #General: The MS needs careful perusal and correction by the authors. Several words seem to me to be computer-generated automatic characters and must be replaced.

For example: Line 157- “stme”- to change to “ stem”, Line 177- “secondary triangles” to change to “secondary sclerotia” and more.

Response: I checked our manuscript to correct misspellings.

Comment #Abstract 1: Please add the aim of the study and methods. In my opinion the abstract contains only the introduction, description of the appearance of a new natural population and conclusion. Please add a description of the lab experiments molecular methods, lab studies on the effect of environmental conditions on growth rate and sclerotia formation etc., as well as the results that enabled your conclusions.

Response: I described the aim and methods in abstract of revised MS (L.31-35). “In this study, we aimed to elucidate the life cycle plasticity in the genera Typhula and Pistillaria though the interactions between their ecophysiological potential and environmental conditions in their localities. We collected and prepared stains of the above fungi from sclerotia or basidiomata, and we elucidated the taxonomical relationship and determined the physiological characteristics of our strains.”

Comment #Abstract 2: In line 24 :“and consequently the view of Karsten (1882) has been widely accepted; Typhula develops basidiomata from sclerotia” please add “:” instead of “;”.

Response: I changed the abstract according to this suggestion.

Comment #Keywords: Please mention the names of the fungi studied: Typhula hyperborea and Pistillaria petasitis

Response: I added two species in keyword according to this suggestion.

Comment #Introduction 1: The introduction is too long and has basic information on the fungal world. It can be made concise by focusing on the topic of the paper.

Response: I removed descriptions of “cryosphere” and “concept of fungi” and cited references in this section.

Comment #Introduction 2: Line 56: Please explain why mycorrhizal fungi have been defined as cryophilic fungi.

Response: Several mycorrhizal fungi were recorded in the cryosphere, and these fungi were called “snowbank fungi”. I described “snowbank fungi” in the revised MS (L. 54) and added this reference [7].

Comment #Introduction 3: Line 60 starts with “Their life cycles”. Whose life cycles? General fungi, or cryophilic fungi? All fungi are affected by environmental factors.

Response: I changed “life cycles of cryophilic fungi” in the revised MS (L. 55).

Comment #Introduction 4: Line 84- “In this study, we aimed to elucidate the plasticity in the genera Typhula and Pistillaria”. Please define the plasticity you aim to elucidate, whether “life cycle plasticity” or another similar sentence.

Response: We are interesting in and focused on “life cycle plasticity”. I changed these sentences in the revised MS (L31 and 79).

Comment #Introduction 5: Line 86- This is summarizing a conclusion that remains to be proved.

Response: I removed following sentence “The results here revealed that repeated basidioma production was dependent on both ambient temperatures and the length of time without snow cover after basidioma formation” from revised MS.

Comment #Material and methods 1: Please add the names of the studied fungi in the first section "Fungal material". I find it confusing that several fungi have been studied, with different names appearing in different sections. I could not find the outline. How many are important and how many are just for comparison?

Response: We collected sclerotia of T. hyperborea (revised MS, L. 84) and basidiomata of Typhula sp. and P. petasitis (revised MS, L. 92).

Comment # Material and methods 2: Molecular identification. Which of the fungi were identified by this method in this paper? All? Only the newly 2 isolated mushrooms? Please add.

Response: We identified T. hyperborea from DNA analysis. Typhula sp. and P. petasitis were identified from morphological characteristics.

Comment #Results: Line 177: Fig 2- Replace the word triangles in "secondary triangles" to "secondary sclerotia".

Response: I changed these words in the text according to this suggestion.

Comment # Results: Line 196- the isolate “ strain OUP1811” appears here for the first time. Please describe the genus, history and reasons for adding it here.

Response: I described previous results of the strain OUP1811 in L. 165-170, the revised MS.

Comment #Results: Figure 4 E. The fungi are not visible. You could change the legend of this figure to fit the picture. It appears in the same place as "D", but zoomed- out.

Response: I changed figure legends 4D and E to “Field observation of heights of basidioma on 3rd August (D) and 23rd August in 2010 (E)”.

Comment #Results: Figure 5 line 232: Please explain why the data are of the month of September, while the mushrooms were collected in August.

Response: We collected Typhula sp. in Ashoro at early October. Therefore, I used monthly air temperature of September.

Comment # Conclusion: This section seems to be in part, a continuation of the result section with more investigations and results. Moreover, the discussion seems to be an extended review. I suggest to shorten this section and to move part of the discussion to another review article.

Response: I removed the descriptions of “Sclerotium nivale” and “ecological strategies of cold-adapted fungi” from the revised MS.

              I believe the manuscript has been improved satisfactorily and hope it will be accepted for publication in Microorganisms.

              Best regards

                                                                                    Sincerely yours,

                                                                                    Tamotsu HOSINO, Ph.D.

Reviewer 2 Report

Tamotsu Hoshino and colleagues present an article on aspects of the life cycle of cryphilic fungi Typhula and Pistillaria.  Specifically, putative interactions with environmental conditions were observed, claiming that production of certain mycelial structures was dependent on both ambient temperatures and the length of time without snow cover.

In general, the article contains some interesting observations for people working in this field, offering more information on these otherwise less described fungal genera, however, in my opinion, the way the data are presented lead to confusion in certain text passages and seem not to follow a specific scientific direction. What is the actual scientific question in this study? The manuscript does not seem to be focused on a specific scientific goal.  In order to highlight novelty and to support the conclusions, authors should consider reorganizing the overall structure of the text  and rewrite specific parts, in such a way to either completely separate the discussion from the results and elaborate more on the results section, or provide a single “Results and Discussion” chapter. In this context, certain figures could be combined and placed closer to the relevant text. 

Line 156: The mentioned phylogenetic analysis is quite relevant and should be shown and discussed.

Line 214-216: These specific environmental conditions of each area should be provided in more detail.

Figure 5: A comparison of growth on petri dishes could also nicely accompany the growth/temperature chart

Minor editing is necessary, for example in Line 157 "stme-like"

Author Response

 Thank you so much for the comments from the two reviewers. I am sending herewith our revised manuscript. Our incorporation of reviewer's suggestions is as follows:

We are grateful to reviewer 2 for the critical comments and useful suggestions that have helped us to improve our paper. As indicated in the responses that follow, we have taken all these comments and suggestions into account in the revised version of our manuscript.

Comment #1 General: What is the actual scientific question in this study? The manuscript does not seem to be focused on a specific scientific goal. In order to highlight the novelty and to support the conclusions, authors should consider reorganizing the overall structure of the text and rewriting specific parts in such a way to either completely separate the discussion from the results and elaborate more on the results section or provide a single “Results and Discussion” chapter. In this context, certain figures could be combined and placed closer to the relevant text.

Response: Our manuscript was focused on the relationship between the climate condition and fungal physiologic characteristics of Typhula and related genera. I revised our manuscript according to your suggestion. I also removed several side-story in the “discussion,” such as “Sclerotium nivale in Russia” and “ecological strategies of cold-adapted fungi” from the revised MS.

Comment #2: Line 156: The mentioned phylogenetic analysis is quite relevant and should be shown and discussed.

Response: I made the phylogenetic tree (Figure S1) and added in the revised MS.

Comment #3: Line 214-216: These specific environmental conditions of each area should be provided in more detail.

Response: Reviewer’s suggestion was described in L. 216-218 (the revised manuscript).

Comment #4: Figure 5: A comparison of growth on petri dishes could also nicely accompany the growth/temperature chart

Response: I am sorry. I did not have suggested photos of mycelial growths on PDA at various temperatures.

              I believe the manuscript has been improved satisfactorily and hope it will be accepted for publication in Microorganisms.

              Best regards

                                                                                    Sincerely yours,

                                                                                    Tamotsu HOSINO, Ph.D.

Round 2

Reviewer 2 Report

The revised manuscript has been improved.

I have no further comments for the authors.